# Reelin Signaling and Synaptic Plasticity in Schizophrenia

**DOI:** 10.3390/brainsci13121704

**Published:** 2023-12-11

**Authors:** Renata Markiewicz, Agnieszka Markiewicz-Gospodarek, Bartosz Borowski, Mateusz Trubalski, Bartosz Łoza

**Affiliations:** 1Occupational Therapy Laboratory, Chair of Nursing Development, Medical University of Lublin, 4 Staszica St., 20-081 Lublin, Poland; renatamarkiewicz@umlub.pl; 2Department of Normal, Clinical and Imaging Anatomy, Medical University of Lublin, 4 Jaczewskiego St., 20-090 Lublin, Poland; 3Students Scientific Association, Department of Normal, Clinical and Imaging Anatomy, Medical University of Lublin, 20-090 Lublin, Poland; bartosz.borowski@interia.pl (B.B.); mateusztrub@gmail.com (M.T.); 4Department of Psychiatry, Medical University of Warsaw, 02-091 Warsaw, Poland; bartosz.loza.med@gmail.com

**Keywords:** reelin, neuronal migration, neurodevelopment, neuroplasticity, synaptic plasticity, schizophrenia, GABAergic interneurons, cortical development malformations, mental disorders

## Abstract

Recent research emphasizes the significance of studying the quality of life of schizophrenia patients, considering the complex nature of the illness. Identifying neuronal markers for early diagnosis and treatment is crucial. Reelin (RELN) stands out among these markers, with genetic studies highlighting its role in mental health. Suppression of RELN expression may contribute to cognitive deficits by limiting dendritic proliferation, affecting neurogenesis, and leading to improper neuronal circuits. Although the physiological function of reelin is not fully understood, it plays a vital role in hippocampal cell stratification and neuroglia formation. This analysis explores reelin’s importance in the nervous system, shedding light on its impact on mental disorders such as schizophrenia, paving the way for innovative therapeutic approaches, and at the same time, raises the following conclusions: increased methylation levels of the *RELN gene* in patients with a diagnosis of schizophrenia results in a multiple decrease in the expression of reelin, and monitoring of this indicator, i.e., methylation levels, can be used to monitor the severity of symptoms in the course of schizophrenia.

## 1. Introduction

This study aims to explore researchers’ findings on the connection between disorders in the reelin signaling pathway and the development of schizophrenia. It seeks to uncover the implications these insights may have on the diagnosis and holistic treatment of schizophrenia.

Reelin, an extracellular matrix glycoprotein, plays a crucial role in neuronal migration and development of the new cortex [1,2]. Its influence extends beyond neurogenesis and cortical specialization, encompassing the sustained activity of the brain in adulthood. Reelin actively regulates glutamatergic neurotransmission and contributes to the formation of dendritic spines, thereby maintaining synaptic plasticity at the optimal level [3,4,5]. The binding of reelin to receptors, such as very low-density lipoprotein receptor (VLDLR) and ApoE receptor 2 (ApoER2—apolipoprotein E gene), triggers the phosphorylation and activation of intracellular disabled protein-1 (Dab-1—1,4 dideoxy-1,4-imido-D-arabinitol). This sets off a complex signaling kinase cascade involving PI3K (phosphatidylinositol 3-kinase), Erk1/2 (extracellular signal-regulated kinase 1/2), and GSK3 (glycogen synthase kinase-3). These pathways are responsible for dendritic proliferation, initiating dendritic growth, and branching in the cerebral cortex, particularly in regions such as the hippocampus [6,7,8,9,10]. While initially hypothesized to function as a serine protease with proteolytic activity driving key functions like migration, adhesion, and neuroplasticity, studies have yielded divergent results [11,12].

Genetic studies have linked reelin to the pathogenesis of neuropsychiatric diseases, such as schizophrenia, bipolar, and autism spectrum disorders [13,14,15,16]. This could be the result of an early error in neurogenesis as well as reduced or suppressed expression of reelin, leading to abnormal neuroplasticity, especially limiting dendritic proliferation. Ectopic dendrites, abnormal circuits, and abnormal neuronal migration are then observed [17,18,19,20,21,22]. This raises the question of how the dysregulation of reelin expression affects the development of neuropsychiatric disorders.

Despite the intricate interplay of circuits, cortical–subcortical loops, and feedback mechanisms—both hierarchical and positive–negative—of which the reelin signaling pathway stands out as a notable example, research on this pathway in the context of schizophrenia has predominantly relied on post-mortem studies, completed by genetic tests [23,24,25,26]. Post-mortem examinations have revealed a decrease in reelin expression within the central nervous system (CNS) among affected individuals, a trend also observed in vivo [26,27,28]. Intriguingly, schizophrenic patients exhibited significantly elevated serum levels of major reelin isoforms [29]. Furthermore, alterations in the activity of reelin-specific receptors, such as very low-density lipoprotein receptor (VLDLR) and apolipoprotein E receptor type 2 (Apo-ER2), were noted during antipsychotic treatment [30,31,32]. Drawing on available studies from PubMed and Scopus, our review delves into publications exploring the dysfunction of the reelin signaling pathway, Cajal–Retzius cells, and GABAergic interneurons. Our focus is on unraveling their roles in the etiopathogenesis, cognitive predispositions, and clinical symptoms of schizophrenia. Keywords such as schizophrenia, synaptic plasticity, reelin, neuroplasticity, and mental disorders guided our literature review, with a time frame limited to the last 10 years. However, we also incorporated older literature to provide a comprehensive understanding of the underlying mechanisms in our exploration.

## 2. Physiological Significance of Reelin in CNS

Reelin, a protein predominantly found within the brain, was first discovered in the 1950s and has since been recognized as a critical regulator of various developmental processes in the CNS. This glycoprotein plays a pivotal role in guiding the migration and proper localization of neurons during embryonic brain development. It is primarily produced by specialized Cajal–Retzius cells situated in the marginal zone of the developing brain [33,34]. These cells are indispensable for orchestrating the migration of ventricular neurons, ultimately contributing to the formation of highly organized laminated structures within brain regions such as the neocortex, cerebellum, and hippocampus [35,36]. Disruption of reelin production during this pivotal developmental stage, or interference with the signal transduction pathways initiated by reelin, can have severe consequences on the morphology of the CNS [37].

While reelin’s initial exploration centered around its role in embryonic CNS development, contemporary research has shifted its focus toward understanding its significance in adult synaptic signaling [38]. Within the cerebellum, numerous cells responsible for producing reelin are strategically located near the surface of the developing cerebellar cortex. These cells can be found in specific regions, including the rhombic lip migratory stream, the nuclear transitory zone, and the external granular layer. Moreover, there are additional reelin-producing cells situated in specific deep cerebellar nuclei and the internal granular layer [38]. In this context, a vital function of reelin is to facilitate the detachment of Purkinje cells from radial glia located in the mantle zone. This detachment process is instrumental in ensuring the smooth migration of Purkinje cells to their ultimate destinations within the cerebellum, highlighting the multifaceted role of reelin in both developmental and adult brain functions [39,40].

Reelin pathway abnormalities have been linked to various neurological disorders, including schizophrenia and neurodegenerative conditions like Alzheimer’s disease [41,42]. A study examining post-mortem brain samples from schizophrenic patients revealed significantly reduced reelin and reelin mRNA expression compared to unaffected individuals, particularly in prefrontal regions (Brodmann’s 10 and 46 areas), the temporal cortex (Brodmann’s 22), the hippocampus, caudate nuclei, and cerebellar tissue [39,43].

More recent studies found increased reelin gene (RELN) methylation in schizophrenic patients compared to controls. This methylation led to a substantial 25-fold decrease in reelin expression in the affected individuals [44].

Reports indicate reelin’s potential role in schizophrenia development, possibly due to its impact on the laminar structure of the cortex and synaptic signaling, which in turn would result in changes in the expression of reelin in the CNS and serum as well as changes in the expression of its receptors during treatment [26,27,28,29,30,31,32,45]. Currently, systematic research is needed to link the clinical effects of treatment with changes in the reelin signaling pathway as the primary target of antipsychotic therapy.

## 3. Reelin Signaling Pathway

### 3.1. The Role of Reelin Protein

As mentioned before, reelin’s main feature is conducting the development and organization of the brain cortexes’ neuronal layers, which is why understanding its signaling pathway is important for researchers [46]. This specific signaling pathway yields correct tissue conformation. In the early stages of life, reelin is mainly produced by Cajal–Retzius cells, while in adulthood, it is produced by GABAergic interneurons [47].

### 3.2. Receptor Binding

Firstly, reelin connects with two extracellular receptors—VLDLR (mostly common in the cerebellum) and ApoER2 (common in the cerebral cortex) [48]. These are lipoprotein-type transmembrane receptors, from the LDLR (low-density lipoprotein receptor) family. It has been discovered that abnormal functionality of these receptors leads to structural and functional disorders of the brain and cerebellum, with more severe abnormalities when both receptors are affected [49,50].

### 3.3. Navigating the Second Path

Later, activated VLDLR and ApoER2 induce phosphorylation of the Dab-1 protein. The Dab-1 mutation also seems to cause similar neuronal disruptions as LDLR receptor abnormalities, which are described as having a “reeler-like phenotype” [48]. According to Ramsden et al. (2023), the ApoER2-Dab-1 disorder might be crucial in the pathogenesis of sporadic Alzheimer’s disease [50]. After Dab-1 activation, a cascade of molecules and enzymes activates, such as C3G (cyanidin 3-glucoside), Akt (serine/threonine kinase 1), Crk (CT10 regulator of kinase), CrkL (Crk-like), Rap 1 small GTPase (guanosine triphosphatase), and others [48]. The described route is presented in Figure 1.

### 3.4. Abnormalities in the Signaling Pathway

Some studies have been conducted to better understand the specific role of reelin in human brain development and, especially, to find out whether its abnormal level or function takes part in neurodegenerative and brain diseases associated with psychotic manifestations. Mutations of RELN or other genes responsible for coding enzymes participating in the reelin signaling pathway might lead to neuropsychiatric disorders [51,52,53,54,55].

### 3.5. Possible Research Regarding the Reelin Signaling Pathway

In animal models, especially rodents, there have been observed emotional, motor, and learning deficits associated with abnormal reelin levels [56,57,58]. This finding encourages researchers to identify whether normalization of the protein’s level might aid disease treatment, and some papers suggest that it might decrease psychopathological symptoms [59,60,61,62]. Moreover, the monitoring of DNA methylation related to reelin expression might be used as a marker in assessing the severity of the symptoms appearing in schizophrenia [62].

Reelin deficiency can change fear learning when comorbid with Alzheimer’s disease risk factors [63]. The fear-learning model, a type of learning when neutral stimuli are paired with aversive ones, is mediated by several metabolic systems, such as the tryptophan–kynurenine pathway [64,65]. It has been demonstrated that disturbances in fear-learning systems may be a common gateway to the development of basic mental disorders [63,64,65]. Both the reelin and tryptophan–kynurenine pathways can be genetically linked and negatively correlated [66].

## 4. Reelin Functions in Neuronal Migration

The expression of reelin from Cajal–Retzius and Cajal–Retzius-like cells in the embryonic forebrain is a decisive factor in the process of neuronal migration and plate layer formation within the developing cerebral cortex [67,68]. During embryonic stages, these specialized cells play a vital role in guiding the migration of newly formed pyramidal neurons from their origin in the ventricular zone towards the outer surface of the cerebral cortex. This migration is facilitated by the presence of radial glial processes, which serve as guiding tracks for the migrating neurons [68,69].

In the cerebellum, reelin-producing cells are primarily located near the surface of the developing cerebellar cortex. These cells are found in various regions, including the migratory stream of the rhombic lip, the nuclear transitory zone, and the external granular layer. Additionally, some reelin-producing cells can be found in deep cerebellar nuclei and the internal granular layer [38,68]. One notable function of reelin in the cerebellum is its involvement in promoting the detachment of Purkinje cells from radial glia in the mantle zone. This detachment process facilitates the migration of Purkinje cells to their final destinations within the cerebellar cortex [38]. In a fascinating inside-out patterning, the developing cerebral cortex undergoes a complex layering phase. As neurons are generated, they populate the cortical plate in a specific sequence, resulting in a distinct six-layered structure. Early-generated neurons tend to accumulate in the deeper layers, while newly arriving cells occupy the more superficial layers, where migration eventually comes to an end [69,70].

At the outermost layer of the developing cortex, known as the marginal zone, neuronal migration is halted; this zone serves as a boundary where migrating neurons stop and establish their final positions. During embryonic life, the marginal zone exhibits the highest level of reelin expression, which contributes to the regulation of neuronal migration; however, following the completion of neuronal migration, the Cajal–Retzius neurons, responsible for reelin production, gradually disappear [38,71].

During the process of radial migration, reelin triggers the activation of Rap1, which subsequently controls the expression of N-cadherin. Neurons located in the intermediate zone of the cortex undergo polarization and concentrate N-cadherin within their cell membranes, forming what is known as the leading process. A decrease in N-cadherin expression results in a delay in both polarization and migration. In the absence of Rap1, pyramidal neurons in the cortex still undergo polarization; however, the stabilization of the leading process and nuclear translocation are disrupted [71,72].

N-cadherin also participates in hemophilic interactions, which play a role in consolidating crucial adhesion points necessary for nuclear translocation. Consequently, the signaling pathway mediated by reelin induces the polarization of neurons within the intermediate zone, causing them to transition from a multipolar to a bipolar morphology. Simultaneously, it stabilizes the leading processes through the regulation of adhesive focal molecules. This process is of utmost importance for facilitating nuclear translocation near the marginal zone [69,70,71,72].

The specific localization of ApoER2/VLDLR is vital in determining the functional role of reelin. Activation of reelin signaling leads to the phosphorylation of cofilin in the leading processes of migrating neurons. However, in mutant mice lacking the reelin gene—RELN (Reeler mutant mice)—as well as in mice with a double knockout of ApoER2 and VLDLR, and the Dab-1 mutant, there is a significant reduction in cofilin phosphorylation. It is suggested that ApoER2 has a greater influence on the reelin-induced phosphorylation of cofilin compared to VLDLR [69,70,71,72,73].

ApoER2 is present predominantly in multipolar cells located in the intermediate zone. On the other hand, VLDLR exhibits primary expression in the cortical plate, specifically near reelin-producing cells situated in the marginal zone. VLDLR is specifically localized in distant regions within the marginal zone. Distinct patterns of receptor expression likely contribute to the diverse effects exerted by reelin on migrating neurons during radial migration in the developing cerebral cortex [72,73]. Upon reaching the reelin-containing marginal zone, the leading processes of neurons establish connections, leading to detachment from the radial glial fibers and transitioning from a migratory state to nuclear translocation. The binding of reelin to alpha3beta1 integrin (CD 49c/29) receptors and its interaction with Dab-1 is mediated by intracellular domains. This interaction plays a role in inducing the detachment of migrating neurons from the radial glia [72,74].

Reelin signaling also plays a role in regulating microtubule dynamics during neuronal migration. Through its binding to ApoER2, reelin inhibits the activity of GSK-3β. GSK-3β is known to phosphorylate tau, a protein involved in stabilizing microtubules. In mutant mice lacking the reelin gene (Reeler mutant mice), there is an increase in GSK-3β activity, leading to the hyperphosphorylation of tau [69,75]. This, in turn, results in the destabilization of the tubulin cytoskeleton. Additionally, Lis1, a subunit of platelet-activating factor acetylhydrolase (PAF-AH), encoded by the Pafah1b1 gene, has been found to interact with reelin signaling and is equally important in the process of neuronal migration [75,76,77].

Disruption of the reelin or Lis1 gene leads to lissencephaly, a neurological disorder characterized by impaired neuronal migration in humans. Lis1 interacts with tubulin, suppressing microtubule dynamics and reducing microtubule catastrophe events, thus promoting microtubule stabilization. In terms of reelin’s involvement in nuclear translocation, it has been observed that reelin facilitates the formation of microtubule plus ends binding protein 3 (EB3) in the marginal zone [72,77,78]. Notably, inhibiting Lis1 function solely blocks nuclear translocation without affecting the centrosome in the leading process. This suggests that the movement of the nucleus and centrosome are regulated differently. Consequently, reelin stabilizes the actin cytoskeleton through its interaction with ApoER2, while it stabilizes the microtubule cytoskeleton by binding to VLDLR, thus promoting nuclear translocation [72,76,77,78,79].

## 5. Reelin Expression and Schizophrenia

### 5.1. Neurodevelopmental Hypothesis

Because reelin regulates the neuroblast cytoarchitecture, including the migration and positioning of pyramidal neurons, interneurons, and Purkinje cells, and, after birth, still modulates synaptic plasticity, it has become one of the obvious candidates for a neurodevelopmental hypothesis of schizophrenia [77,78,79,80,81,82]. Importantly, in the neurodevelopmental hypothesis, the disease is the result of the confluence of various factors, biological and psychosocial, acting sequentially at two critical milestones (two-hit model), i.e., initial early damage at the end of the first or second trimester of pregnancy, e.g., as a result of viral infection or psychosocial stress, and then the clinical manifestation of the disease in early adulthood, cumulatively after the final phase of puberty in the presence of individual predispositions and modifying factors, e.g., genetic, familial, psychoactive drug-related or others [81,82]. A decrease in reelin expression might be a crucial trigger or vulnerability factor in developing schizophrenia [26].

### 5.2. Reelin Expression

In post-mortem studies, schizophrenia patients showed a half-reduced concentration of reelin and reelin mRNA in the prefrontal and temporal cortex, as well as in the hippocampus, caudate nucleus, and cerebellum [26]. However, no significant change was found in the same regions for Dab-1 (mouse-disabled-1), a key regulator of reelin intracellular signaling whose mutations generate identical phenotypes as reelin deficit. The effect did not depend on the key schizophrenia distinction: paranoid or undifferentiated clinical forms. The reelin decrease was coupled with reduced expression of GAD-67 enzyme (glutamic acid decarboxylase), which catalyzes the transition of glutamate to GABA (gamma-aminobutyric acid). The same outcomes were confirmed not only in schizophrenia but also in bipolar disorder patients when psychotic symptoms were present, although not in unipolar depression patients [45]. No relation to the characteristics of antipsychotic treatment was demonstrated [45]. Similarly, engaging several research centers and laboratories, a significant decrease in the expression of reelin mRNA was once again confirmed in the prefrontal cortex of post-mortem patients with schizophrenia [83]. A down-regulation of reelin expression occurring in the prefrontal cortex of schizophrenia patients may be associated with a decrease in dendritic spine plasticity, which is probably related to ineffective reelin–integrin interactions [84].

During in vivo studies, pluripotent neurosphere-derived olfactory cells were generated from the nasal mucosa biopsies of schizophrenia patients [27,28]. Levels of reelin mRNA and reelin protein were reduced. Growing in vitro, olfactory cells were unable to respond to reelin and alter their goal migration [27,85].

### 5.3. Epigenetic Hypermethylation

The reduction of reelin and GAD-67 in the prefrontal neurons of schizophrenia brains seems to be the consequence of the epigenetic hypermethylation of their promoters by the overexpression of DNA-methyltransferase 1 (DNMT1) [86]. In a way, this alludes to experiments from more than half a century ago, when a high dose of amino acid l-methionine was administered to patients with schizophrenia, causing significant clinical exacerbations [87]. Valproate, which is the histone deacetylase inhibitor, was able to prevent l-methionine-induced reelin and GAD-67 promoter hypermethylation in animal studies [86].

A mouse experiment in which reelin and GAD-67 promoters suffer from hypermethylation, leading to the formation of transcriptionally inactive chromatin, was reversible thanks to the administration of histone deacetylase inhibitors, such as valproate [85]. This may be potentially important for confirming the methylation/demethylation hypothesis of schizophrenia and for designing antipsychotic treatment [85]. In another study, rats were pre-treated with cortisol and then reelin was directly administered to the hippocampal region. A single-reelin infusion increased the number but not the complexity of newborn neurons; however, a repeated reelin infusion restored both their number and complexity [88].

A two-fold reduction in both GAD-67 and reelin mRNA in GABAergic interneurons isolated from layer I of the cerebral cortex of patients with schizophrenia could be the result of a three-fold increase in DNMT1 activity [89]. This effect was site-specific (layer I, not layer V) and cell-specific (reelin-secreting GABAergic, not glutamatergic neurons). A two-fold increase in the concentration of the methyl donor S-adenosyl methionine (SAM) was found in the prefrontal cortex of schizophrenia and bipolar disorder patients, but not in unipolar depressed ones [90]. Although SAM seems to be a key epigenetic regulatory factor as the methyl donor in cytosine methylation, and the first attempts to use it directly in the treatment of schizophrenia took place half a century ago [91], the results of meta-analyses are inconclusive [92], and the use of SAM itself can bring paradoxical biochemical effects and serious adverse effects [93].

An increased blood reelin protein concentration was found in schizophrenic patients with a first psychotic episode of the paranoid type in comparison to healthy controls [94]. The relationship between the DNA methylation level of reelin promoters and schizophrenia, derived from the peripheral blood, proved a significantly higher level of methylation compared to controls, and in males compared with females [62].

### 5.4. Reelin and Its Circulating Isoforms

Upon proteolytic cleavage between reelin repeats 2 and 3 as well as between repeats 6 and 7, the reelin molecule can be divided in various ways, but only those isoforms that contain the central fragment R3–R6, and especially those containing the R5 and R6 repeats, have activity within the reelin–Dab-1 signaling pathway (Figure 2) [40,47].

The formation of a reelin N-t cleavage site appears to be crucial for the CNS, and in an experiment in which this cleavage was abolished in mice, the hippocampal layers were disturbed [95]. At the same time, the dendrites of these mutant mice had more branches and were elongated compared to wild-type mice. N-t cleavage requires many enzymes for maturation (from the protease and convertase families) [34]. Splitting reelin in this way virtually abolishes its signaling activity [34].

In addition to the N-t and C-t cleavage sites, there is also the less significant reelin WC cleavage site, which occurs between Arg3455 and Ser3456, only six amino acids from the C terminus [95].

Since reelin is a secreted extracellular matrix protein, measurements of its serum isoforms have been performed for various psychiatric disorders, including schizophrenia [29]. The serum levels of reelin 410 kDa and 330 kDa, but not 180 kDa, were significantly increased in schizophrenic patients, contrary to bipolar and unipolar mood disorders vs. normal controls [29]. Regarding the 410 kDa fragment, only the results in schizophrenic patients were significantly different (increased) vs. the control group. For the 330 kDa fragment, those with schizophrenia had significantly elevated levels, those with bipolar disorder had decreased levels, and those with unipolar disorder did not present significant changes. As for the 180 kDa fraction, levels were not significantly changed in people with schizophrenia but were significantly decreased in people with unipolar and bipolar disorders.

Reelin 180-kDa is increased in frontotemporal dementia, progressive supranuclear palsy, and Parkinson’s disease, and is positively correlated with tau levels in CSF (cerebrospinal fluid) [96]. The study of specific isoforms of reelin in neurodegenerative diseases may be more specific diagnostically than the general measurement of reelin in CSF [97].

Mothers who were exposed during pregnancy to selective serotonin reuptake inhibitors (SSRI class antidepressants) had higher levels of 310 kDa reelin, while their female neonates had lower 310 kDa levels, which was associated with less time spent sleeping and more irritable behavior in 6-day-old newborns [98].

Changes in the levels of reelin in some somatic diseases have also been observed. Reelin was increased in the synovial fluid of rheumatoid arthritis patients and the serum of HCV liver fibrosis patients, and 180 kDa reelin was increased in the plasma of liver cirrhosis patients [99,100]. Heterozygous reelin mutations can cause autosomal-dominant temporal epilepsy with lower serum levels of reelin [101]. A reduced blood level of reelin is a vulnerability factor in the pathophysiology of autistic spectrum disorders [102].

### 5.5. Reelin: Whole-Body Balance

There is no conclusion about the interaction between peripheral and central reelin levels [94]. As any neuroinflammation causes an increase in reelin levels in the serum, it is also produced locally, and its largest source outside the CNS is the liver, especially hepatic stellate cells [47]. The liver appears to be a prime candidate for balancing the circulating reelin pool [103]; the adult liver expresses one third of the reelin mRNA in the cerebral cortex of adult mice [103].

Reelin, as a guidance molecule, may play an unfavorable role in the pathogenesis of atherosclerosis, arthritis, and the development of cancer [47].

Reelin serum levels display significant variations over 24 h and substantial divergence between individuals [29]. In general, the profile gradually and slightly decreases during the day in healthy individuals. Although glucocorticoids can inhibit hepatic reelin synthesis in vitro, no clear decline is observed in serum reelin in opposition to cortisol in humans [29].

### 5.6. Unlocking the Antipsychotic Potential of Reelin

The long-term administration of both typical (haloperidol and flupentixol) and atypical (clozapine and olanzapine) antipsychotics can modify the levels of reelin, its receptor VLDR, and the downstream signaling molecules GSK3 beta, Dab-1, and GAD65/67 in the rat prefrontal cortex [104]. Examination of peripheral blood lymphocytes in patients with schizophrenia expressed low activity of very low-density lipoprotein receptor (VLDLR) for reelin in drug-naive, unmedicated patients, and unchanged activity of apolipoprotein E receptor type 2 (ApoER2) [105]. Levels of VLDLR mRNA in drug-naive patients increased after six months of antipsychotic treatment, whereas ApoER2 mRNA decreased [30]. In another study, the expression of reelin mRNA significantly increased in the peripheral blood of schizophrenia patients after 12 weeks of treatment [106].

Given the reduced levels of reelin observed in the prefrontal cortex of schizophrenia patients, an investigation was conducted to validate reelin’s potential protective role in schizophrenia. Two sets of mice, one with reduced reelin levels and the other with increased levels, underwent behavioral tests [62]. Interestingly, mice exhibiting reelin overexpression outperformed their counterparts in tasks, highlighting the significance of the reelin signaling pathway as a promising target for schizophrenia treatment.

### 5.7. Exploring Alternative Splicing

Reelin transduction occurs in two forms: as a brain-specific alternative splicing event with a microexon of six nucleotides, retained in 83.3% of all post-mortem brain transcripts, and as a product of alternative polyadenylation leading to a truncated protein, accounting for, on average, 7.8% of transcripts [13]. The genotype-by-sex effect was observed in women with the risk genotype of rs7341475 (GG) and, simultaneously, a higher proportion of microexon skipping, which is the dominant form in tissues outside the brain [13].

In studies involving animals, the significance of alternative splicing in various components of the reelin signaling pathway has been highlighted for its overall impact. Alterations in ApoER2 alternative splicing, for instance, have been shown to influence the outcomes of learning and memory tasks [107]. Additionally, the splicing variations in Dab-1 contribute to changes in its phosphorylation potential across different stages of neural development [108].

### 5.8. Unraveling the Genetic Connection: Reelin and Predisposition to Schizophrenia

Genetic studies indicate the relationship between the predisposition to develop schizophrenia and the metabolism of reelin, and, at the same time, at least in part, this significantly depends on the gender of the patient (with higher risk for females) [109,110]. Allelic variants of reelin contributed to the endophenotypes of schizophrenia, and positive subjects scored lower in working memory, memory, and executive functioning [110].

Proteins of virtually the entire reelin metabolic pathway have been linked with genetic studies to transcription factors including NPAS1 (Neuronal PAS 1) [111], NPAS3 (Neuronal PAS domain protein 3) [112], and TBR1 (T-box brain transcription factor 1) [113]; enzymes such as MTHFR (5,10-methylenetetrahydrofolate reductase) [114]; and receptors such as NMDA (N-methyl-D-aspartate) [115], which are considered as gene candidates for schizophrenia. However, in one study, none of 83 SNP (single nucleotide polymorphism) associations with variations in the expression of reelin were significant [13]. In a meta-analysis of reelin polymorphism studies, the results proved controversial, as rs7341475 was associated with decreased schizophrenia risk and rs262355 with increased risk [116].

The modulatory role of reelin in the adult brain can be expressed via alternative splicing of ApoER2, an event that is not required for early neuronal migration [107]. Given that the alternative splicing of ApoER2 may be the basis for synapse formation, more effective sensory input, the consolidation of memory, and vice versa, it may make an adverse contribution to the etiopathology of schizophrenia [107].

## 6. Conclusions and Future Directions

The research outlined above appears to be accurate and reliable, yet it remains inconclusive. This is largely due to the absence of a universally accepted animal model of schizophrenia and a singular model for the human schizophrenia process. These models differ fundamentally in their premorbid, early, and chronic stages; acute and long-term treatment; and in the perception of patients themselves compared to those around them [53].

The physiological and pathological influences of reelin on the CNS formation seem to be well established, while its clinical significance and therapeutic implications deserve further investigation. The key findings from this research are as follows:Disruption of the signal transduction pathway, initiated by reelin, has a detrimental impact on the morphology of the cortical areas of the brain [13,14,15,16,17,18,19,20,21,22,26,50,51,52,53,54,72,80,81,82,84,116].The reelin levels in the CNS associated with neuropsychiatric disorders are decreased [26,27,38,52,53,54,55,97,116]. Contrary to this, the reelin levels in neuropsychiatric disorders may be increased in the serum [30,102] and in the CSF [30,100].The abnormal function of VLDLR and ApoER2 receptors leads to structural and functional disorders of the brain and cerebellum [31,48,49,105].Mutations of the RELN gene or genes encoding enzymes involved in the reelin signaling cascade lead to neurological and psychiatric disorders [13,40,51,52,53,54,107,108].No single genetic variant within the reelin pathway is clearly associated with the development of schizophrenia [13,105,107,108,114].An increased level of methylation of the RELN gene in patients diagnosed with schizophrenia causes a multiple decrease in reelin expression [43,89,90]. Monitoring the level of methylation can be used as a marker in assessing the severity of symptoms occurring in schizophrenia [43,62,89,90,91,92,93].Decreased levels of reelin are associated with reduced expression of the GAD-67 enzyme, which catalyzes the conversion of glutamate to GABA [26,82,86,88].Post-mortem examinations of brain samples of individuals diagnosed with schizophrenia confirm reduced expression of reelin protein and mRNA in the prefrontal and temporal cortices as well as in the hippocampus, caudate nucleus, and cerebellum [38,40,82,83,84].The long-term administration of both typical and atypical antipsychotics can modify the levels of reelin, its VLDR receptor, and downstream signaling molecules [32,45,86,102].To our knowledge, no systematic studies have been conducted so far linking the clinical effects of treatment with changes in the reelin signaling system as the primary target of antipsychotic therapy.

To achieve the goal of clinical research, i.e., the more effective diagnosis and treatment of patients with schizophrenia, the main need is to confirm the basic link between clinical symptoms and the effects of antipsychotic treatment in relation to reelin downstream signaling molecules, identifying their roles as specific markers of psychosis [29,115,116]. The reelin system remains a challenge as a target for the development of new diagnostic, pharmacological, and rehabilitation strategies.

## Figures and Tables

**Figure 1 brainsci-13-01704-f001:**
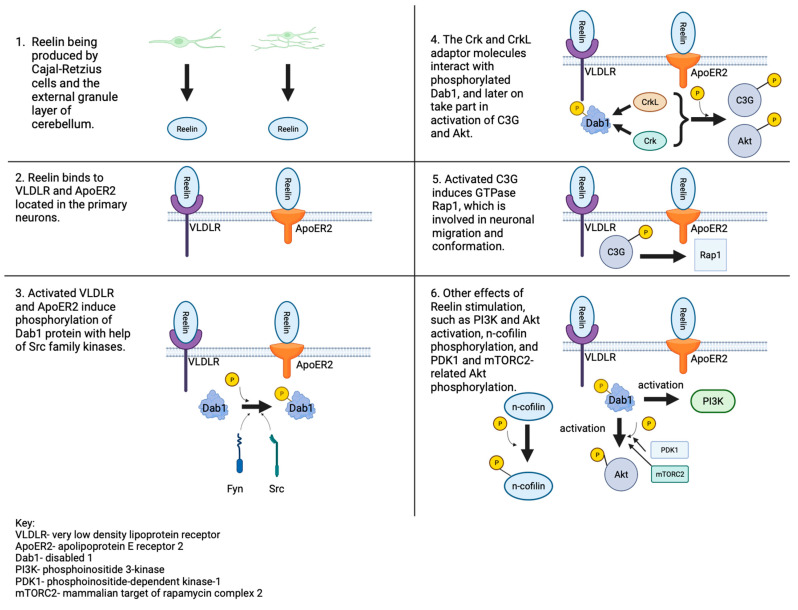
Reelin signaling pathway. The secretion of the protein, its binding to the receptors, and the occurring changes in the neurons [47,49].

**Figure 2 brainsci-13-01704-f002:**
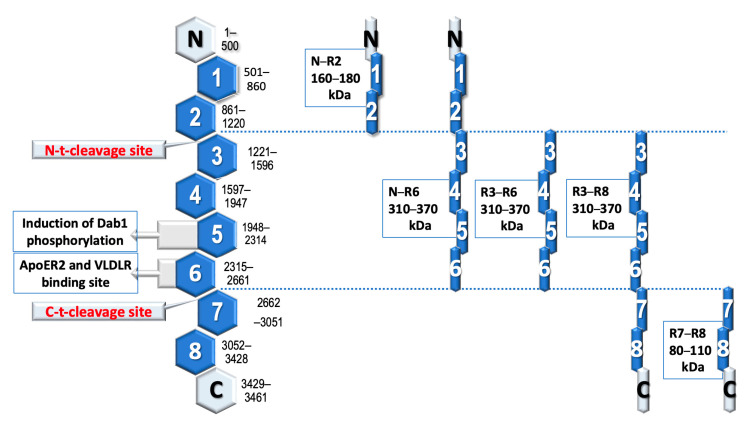
Reelin molecule and its isoforms with reference to amino acid structure, internal repeats, and main cleavage sites [40,47].

## Data Availability

No applicable.

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
