# Peer review of "Reelin Signaling and Synaptic Plasticity in Schizophrenia"

_brainsci, 2023, doi:10.3390/brainsci13121704_

Round 1
Reviewer 1 Report
Comments and Suggestions for Authors
The manuscript provides a comprehensive overview of the role of reelin signaling in the development of schizophrenia, focusing on neurodevelopmental aspects, synaptic plasticity, and potential therapeutic implications.
ABSTRACT
The abstract should be concise and to the point. Try to eliminate redundant phrases and focus on the key findings or insights of the study.
Briefly mention the potential implications of your findings, especially regarding the treatment of schizophrenia. How might this work influence future research or clinical practice?
INTRODUCTION
1. Narrative reviews should still follow internationally accepted guidelines. Consider reviewing SANRA guidelines and add a statement at the end of the introduction stating these guidelines were followed. Adds more credibility to the review.
MAIN REVIEW
The authors mention divergent results in studies exploring reelin as a serine protease. It would be beneficial to provide a more detailed discussion of these contradictions, potentially offering hypotheses or explanations for the discrepancies.
The manuscript does an excellent job of relating reelin's role to schizophrenia's pathogenesis and potential treatments. However, it would be valuable to discuss more explicitly the translational aspects – how can this knowledge be applied in clinical settings? What are the specific future research directions that can emerge from this understanding?
Comments on the Quality of English LanguageThe manuscript is technical and would be well-understood by specialists in the field. However, for a broader readership, including a simplified summary or explanation of complex concepts might be helpful.
Author Response
Dear Reviewer,
Thank You very much for reviewing our manuscript. We appreciate the interest and commitment You have provided for this work. We are very grateful for your extremely precious comments. We are convinced that thanks to Your suggestions this manuscript will be much more valuable.
We are pleased to submit explanations and details of our revisions in the manuscript entitled “Reelin signaling and synaptic plasticity in schizophrenia”.

Reviewer 2 Report
Comments and Suggestions for Authors
The review by Markiewicz and colleagues is of high-interest for the field. The manuscript is generally well written.
Please find below some concerns/suggestions to be addressed:
- It sounds odd reading a very detailed description of Reelin-pathway in the general introduction, while a few words in the dedicated sections 3.1 and 3.2. Maybe the intro can be shortened and some info can be moved under the specific issue.
- The figure 1 is very clear and helpfulf. But several acronyms/abbraviations are missing in the legend.
- I would avoid the name “psychotic diseases” in line 148. Terms like “schizophrenia” or “brain diseases associated with psychotic manifestations” sounds more professional.
- The sentence in line 90-91 sounds misleading, given that in lines 349-368 reelin is well described in the periphery.
The same lines 349-368 can be moved from the dedicated section 5 about schizophrenia, to another general section.
- Despite concerns about animal models for psychiatric disorder are agreeable, writing that reserch “remains inconclusive” (line 421) sound ungenerous, and reserch should be valorised.
- Similarly I would suggest to underlie the importance of research about reelin in line 427-428, for example “clinical significance and therapeutic implications deserve further investigation” instead of “have not yet been confirmed”
Comments on the Quality of English LanguageEnglish is easy readable. I would suggest to split a few long sentences.
Author Response

(The authors gave the same response as above.)
